# Modifiers of Autosomal Dominant Polycystic Kidney Disease Severity: The Role of *PKD1* Hypomorphic Alleles

**DOI:** 10.3390/genes14061230

**Published:** 2023-06-07

**Authors:** Enrico Ambrosini, Francesca Montanari, Carlotta Pia Cristalli, Irene Capelli, Claudio La Scola, Andrea Pasini, Claudio Graziano

**Affiliations:** 1Medical Genetics Unit, University of Parma, 43126 Parma, Italy; enrico.ambrosini@unipr.it; 2Medical Genetics Unit, IRCCS Sant’Orsola University Hospital of Bologna, 40138 Bologna, Italy; francesca.montanari2@studio.unibo.it (F.M.); carlotta.cristalli2@unibo.it (C.P.C.); 3Nephrology Unit, IRCCS Sant’Orsola University Hospital of Bologna, 40138 Bologna, Italy; irene.capelli4@unibo.it; 4Paediatric Nephrology Program, Paediatrics Unit, IRCCS Sant’Orsola University Hospital of Bologna, 40138 Bologna, Italy; claudio.lascola@aosp.bo.it (C.L.S.); andrea.pasini@aosp.bo.it (A.P.); 5Medical Genetics Unit, AUSL Romagna, 47522 Cesena, Italy

**Keywords:** ADPKD, hypomorphic variants, disease modifiers, biallelic inheritance, *PKD1*, *PKD2*

## Abstract

Autosomal dominant polycystic kidney disease (ADPKD) is the most common genetic cause of kidney failure in adult life. Rarely, ADPKD can be diagnosed in utero or in infancy, and the genetic mechanism underlying such severe presentation has been shown to be related to reduced gene dosage. Biallelic *PKD1* variants are often identified in early onset ADPKD, with one main pathogenic variant and a modifier hypomorphic variant showing an in trans configuration. We describe two unrelated individuals with early onset cystic kidney disease and unaffected parents, where a combination of next-generation sequencing of cystic genes including *PKHD1*, *HNF1B* and *PKD1* allowed the identification of biallelic *PKD1* variants. Furthermore, we review the medical literature in order to report likely *PKD1* hypomorphic variants reported to date and estimate a minimal allele frequency of 1/130 for this category of variants taken as a group. This figure could help to orient genetic counseling, although the interpretation and the real clinical impact of rare *PKD1* missense variants, especially if previously unreported, remain challenging.

## 1. Introduction

Autosomal dominant polycystic kidney disease (ADPKD) is the most common hereditary form of renal disease and one of the most common genetic disorders in humans, with an estimated prevalence at birth of 1:1000 [1]. It is the fourth leading cause of end-stage renal disease (ESRD) worldwide, but the clinical course varies considerably among individuals, with some reaching ESRD before 40 years of age and others living a normal lifespan without requiring renal replacement therapy [2]. The two major genes involved are *PKD1* (chromosome locus 16p13.3), responsible for ~78% of cases, and *PKD2* (4q21), responsible for ~15% of cases [1,3]. Genotype–phenotype correlations are well established: pathogenic variants in *PKD1* are associated with more severe disease with an earlier age at diagnosis than in *PKD2*-related ADPKD [4], and truncating *PKD1* mutations lead to earlier ESRD rather than non-truncating variants [5]. 

Although these two genes classically show an autosomal dominant inheritance pattern and adult onset of disease, biallelic variants have been described, especially in early (childhood) or very early (prenatal) onset cases [6]. In this group of patients, earlier onset with respect to an affected parent can be caused by coinheritance of the familial inactivating *PKD1* mutation plus an additional low-penetrant variant on the second allele [7,8,9,10]. Furthermore, ADPKD can be caused by a de novo variant in sporadic cases, but some patients were found to carry biallelic *PKD1* variants, inherited from healthy or mildly affected parents; these variants acted as hypomorphic or reduced-penetrance alleles [11]. Occasionally, digenic inheritance has also been described, with patients carrying a trans-heterozygous pathogenic allele in *PKD2* and a hypomorphic variant in *PKD1* [12,13,14,15]. 

According to the established loss-of-function mechanism, due to a dosage/threshold model or somatic second-hit mechanism [16], biallelic full inactivation of both copies of ADPKD genes are unnecessary to cystogenesis as well as incompatible with life, while residual function of low-penetrance alleles can trigger cyst initiation below a critical threshold (approximately 20–30%) [17]. 

Nevertheless, the definition of low-penetrant variants is a major problem: regarding *PKD1*, 30–35% of pathogenic alleles are non-truncating [16], and it is challenging to define pathogenic or neutral changes and their degree of penetrance, also due to the lack of available functional tests. Moreover, some of these hypomorphic alleles are relatively frequent in the general population. For these reasons, most of them are classified as Variants of Unknown Significance (VoUS) or Likely Benign according to ACMG classification [11,18].

Here, we describe two unrelated children with early/very early onset cystic nephropathy and biallelic *PKD1* hypomorphic variants. Inspired by this, we performed an extensive literature review in order to gather available information on biallelic inheritance of *PKD1* hypomorphic variants and applied stringent selection criteria to evaluate their global frequency.

## 2. Materials and Methods

### 2.1. Patient Recruitment

The two probands described in this study were referred to the Paediatric Nephrology Program of the Paediatrics Unit of our hospital and afterwards to the Medical Genetics Unit. Genetic counselling was provided before and after genetic testing, and informed consent was signed by the parents. Ethical review and approval were waived for this study because, according to the local policy, informed consent is considered sufficient for reports of an observational nature concerning a limited number of patients.

### 2.2. Genetic Analysis

Genetic testing was performed on DNA isolated from EDTA peripheral blood using a semi-automatic Maxwell 16 instrument (Promega Corporation, Madison, WI, USA).

Considering the early age of onset in both probands, targeted NGS analysis of the coding sequence of *PKHD1* (MIM#606702; HGNC:9016; RefSeq NM_138694.3)*, DZIP1L* (MIM#617570; HGNC:26551; RefSeq NM_173543.3) and *HNF1B* (MIM#189907; HGNC:11630 RefSeq NM_000458.4) was performed as a first-line genetic exam. Following negative results, genetic testing was focused on *PKD1* (MIM#601313 HGNC:9008, RefSeq NM_001009944.3) and *PKD2* (MIM#173910; HGNC:9009; RefSeqNM_000297.4)*,* using targeted NGS-based testing strategies validated in a diagnostic setting in order to overcome the well-known difficulties related to the molecular characteristic of *PKD1* [19]. Our workflow included LR-PCR targeted re-sequencing for both *PKD1* and *PKD2*, combined with a specific alignment pipeline for *PKD1*; the complete procedure is described in Mantovani et al. 2020 [12]. The raw sequencing data were transferred to the TorrentServer where Torrent SuiteTM performed alignment to a reference genome in order to generate Fastq files, a Binary Alignment Map (BAM) in conjunction with the Binary Alignment Index (BAI) and Variant Call Format (VCF) files. *PKD1* reads were aligned against a modified reference genome based on chromosome 16 of Human Genome 38 (Grch38), where all the nucleotides outside the *PKD1* locus were masked and replaced with “Ns.” All the VCF files were uploaded into Ion Reporter software (Thermo Fisher Scientific Inc., Waltham, MA, USA), selecting the Annotation Variant workflow in order to associate to each variant the nucleotide change in mRNA transcript, the aminoacidic change, the exons or IVSs, and the function. The BAM/BAI files, generated following alignment, were visualized using Integrative Genome Viewer (IGV) software. IGV was used to assess the depth of coverage of the sequencing reads, zygosity, quality of the sequencing reads and the mapping quality. 

Large rearrangements were excluded by multiplex-ligation probe amplification (MLPA). In both families, an EDTA peripheral blood sample from each parent was taken in order to perform segregation studies.

Variant filtering based on population frequency was performed using population databases ExAC, gnomAD [20], 1000Genomes [21] and dbSNP to include only alleles with a minor allele frequency (MAF) ≤0.01. The variants were then annotated according to the guidelines published by the Human Genome Variation Society [22] and classified into five categories, according to American College of Medical Genetics and Genomics (ACMG) standards [18]. To achieve this, we made use of public variant databases (ADPKD Variant Database [23], ClinVar [24] and LOVD [25]) and online tools such as VarSome Premium [26] and Franklin [27].

### 2.3. Literature Review

In order to assess prevalence in the general population and the alleged role in a clinical setting, we conducted a literature review and collected the hypomorphic *PKD1* variants published so far. The inclusion criteria were (1) a confirmed “in trans” configuration of the *PKD1* variants and (2) a known clinical and family history of disease that allowed us to assess the clinical course as undoubtedly more severe in the proband than in the affected parent (or both parents with no renal disease).

Reports in which these data were unavailable were discarded. In order not to overestimate the true frequency of hypomorphic variants, we also excluded patients with a de novo *PKD1* variant in which the phase was not known. Some studies also reported biallelic *PKD2* variants or digenic *PKD1/PKD2* inheritance [12,13,14]: for the sake of this article, we decided to exclude these patients, focusing on *PKD1* variants only.

## 3. Results

### 3.1. Family F1

The proband (F1, III-2, Figure 1a) was the only child of non-consanguineous healthy parents. During her pregnancy, increased nuchal translucency was identified, and cytogenetic analysis on amniocytes revealed a normal female karyotype. Subsequent ultrasound (US) scans did not detect any other malformations. A few days after birth, abdominal US revealed enlarged and hyperechogenic kidneys with no corticomedullary differentiation, without any liver involvement. She was therefore addressed to our pediatric nephrology service and received periodical clinical and instrumental follow-up. No extrarenal manifestations were detected at the time of our report. At 12 years old (last clinical evaluation), she was normotensive and had normal estimated renal function (GFR 187 mL/min/1.73 m^2^) and urinalysis; her kidneys showed bilateral diffuse hyperechogenicity with multiple small cysts (Figure 1b). Family history was unremarkable except for a maternal uncle affected with an unspecified nephropathy. Her mother and father underwent an abdominal US study, at the ages of 44 and 41 years, respectively, and no renal cysts or other structural anomalies were detected. Further imaging (abdominal MRI or CT scans) was not performed in the parents. As informed consent was obtained, a blood sample was collected from the proband. Sequencing and MLPA analysis of *PKHD1* and *HNF1B* were performed as a first basis but resulted negative. Afterwards, analysis of ADPKD genes revealed two previously unreported *PKD1* heterozygous variants: p.Ala3571_Val3572dup and p.Arg3152Trp, whose configuration “in trans” was subsequently confirmed by parental segregation analysis (Table 1). The first variant is an extremely rare in-frame insertion located in a mutational hotspot, classified as likely pathogenic in VarSome and as VoUs in Franklin. The second one is a rare missense variant, an alternative to another likely pathogenic variant in the same codon (p.Arg3152Ser), classified as likely pathogenic in VarSome and VoUS in Franklin. No further genes were tested.

### 3.2. Family F2

The proband (F2, IV-3, Figure 2a) was the second child of non-consanguineous healthy parents. During her pregnancy, her mother was referred to our Genetic Clinic because a moderate risk for trisomy 21 was identified on first trimester non-invasive maternal screening: a normal fetal female karyotype was detected on a chorionic villus sample. She came back to our attention after a 23-week US scan showed bilateral dysplastic hyperechogenic kidneys, with normal amniotic fluid volume and without any additional malformations. No further tests were performed. The amniotic fluid index remained normal throughout pregnancy. She was born at term by uncomplicated eutocic delivery. Soon after birth, abdomen ultrasound revealed enlarged and hyperechogenic kidneys with multiple cystic spots and no hepatic structural anomalies (Figure 2b); serum creatinine and urinalysis were normal. From neonatal age on, she has been followed up at the pediatric nephrology service of our hospital and received clinical and instrumental periodic evaluations. She had no extra renal anomalies except for segmental facial vitiligo. The most recent abdominal ultrasound, performed at the age of 7 years, showed renal cortical hyperechogenicity with multiple cystic lesions ranging from a point-like size to 3–4 mm in diameter, mild right pelvic dilatation (4–5 mm) and normal hepatic echogenicity (Figure 2c). Blood pressure, serum creatinine and urinalysis were within the normal range according to her age.

After postnatal genetic counselling, informed consent was obtained and a blood sample was collected from the proband. Sequencing and MLPA analysis of *PKHD1* and *HNF1B* were performed as a first basis but resulted negative. Afterwards, analysis of ADPKD genes revealed two previously unreported *PKD1* heterozygous variants: p.Ala2042Thr, inherited from her mother, and p.Pro786Thr, inherited from her father. The first missense variant is extremely rare, an alternative to another likely pathogenic variant in the same codon (p.Ala2042Pro), classified as likely pathogenic in VarSome and as VoUS in Franklin. The second one, though absent in general population databases, is poorly conserved evolutionarily and has weak in silico prediction scores, so it is classified as likely benign in VarSome and VoUS in Franklin. In this case, sequencing and MLPA analysis of *NPHP1* was also performed, together with the sequence analysis of other rarer genes related to nephronophthisis (*INVS*, *NPHP3*, *NPHP4, IQCB1*): no pathogenic variant was found in these genes.

As soon as a fetal kidney dysplasia diagnosis was made, her parents underwent abdominal US studies that did not reveal any anomalies. Further imaging (abdominal MRI or CT scans) was not performed in the parents. The mother reported to have relatives affected with ADPKD (see pedigree), but it was not known if genetic tests were ever performed. The proband had two siblings, aged 8 and 3 years old at the time of the report, who were in good general health but did not receive a renal US study or genetic tests (as a parental choice).

### 3.3. Literature Review

Following the criteria mentioned in the Materials and Methods, we selected a total of 50 patients described in the literature: altogether, they harbored 43 distinct hypomorphic variants. The complete list is reported in Table 2 and Table 3. Table 2 lists 14 patients with biallelic hypomorphic variants, with parents described as unaffected or having mild manifestations of disease (e.g., a few kidney cysts or liver cysts only). Table 3 reports 36 individuals who carry a pathogenic and hypomorphic variant in trans configuration: for each selected patient, we first reported the pathogenic *PKD1* variant considered as “main”, together with the parental allele in which it was identified. For each hypomorphic variant, we reported the parental origin, global minor allele frequency (from GnomAD) and ACMG current classification according to the online tools cited above. Finally, we reported the presence of family history related to the disease, including mild cases, the reported age of onset in the proband and main clinical features in the proband. The global list is not exhaustive, and several putative hypomorphic variants from the referenced papers were excluded. The main reasons for exclusions were uncertainty on the phase of the variants (e.g., one of the variants was de novo) and insufficient clinical data. For example, the recent paper from Durkie et al. [11] reported 21 cases of early onset/very early onset ADPKD with possible biallelic inheritance: three cases were discarded since at least one of the variants was in *PKD2*, another three because it was impossible to determine the phase of the variants and, finally, one for insufficient information about family history and inheritance, leaving only 14 cases.

Some variants were reported in more than one study: p.Arg3277Cys (eight reports), p.Asp1332Asn (four reports), p.Arg4154Cys (three reports) and Ile3167Phe (two reports). The most frequent hypomorphic variants in the general population were p.Leu2696Arg, p.Ile3167Phe and p.Arg4154Cys, all with an MAF of nearly 0.001. Nine variants were absent in the general population. Globally, by summing up the MAF of all the putative hypomorphic variants that we could identify, we estimated a minimal allele frequency of 0.0076 (1/130) for *PKD1* hypomorphic variants taken as a group.

In 12 cases (24%), there was no family history of kidney cystic disease, with another 2 cases with very few signs (single cysts) or incomplete penetrance in the family. We also included two patients with de novo variants, since they were reported “in trans” in the respective paper [11].

Variant p.Phe2132Cys is actually described in other two articles, and some authors hypothesized a correlation with Congenital Hepatic Fibrosis [28,29]; they were excluded since they did not meet the selection criteria (e.g., insufficient information).

**Table 2 genes-14-01230-t002:** Results of the literature review for biallelic hypomorphic *PKD1* variants. Recurrent hypomorphic variants are in gray boxes. For selection criteria, see “Materials and Methods”.

Reference	Paternal Allele	ACMG Class	H-MAF (GnomAD)	Maternal Allele	ACMG Class	H-MAF (GnomAD)	Family History	Clinical Data of the Proband
Mantovani 2020 [12]	R459P	3	0	G1185D	3	0	Some cysts in the father	Prenatal US diagnosis, termination of pregnancy
Gulati 2023 [30]	V1971M	3	0.00007	T2250M	2	0.002	Mild cystic disease in the mother	Perinatal onset (1 y) + stillborn
Al-Hamed 2019 [31]	G2713R	3	0	G2713R	3	0	None	Prenatal US diagnosis
Bergmann 2011 [8]	V1274M	3	0.000004	V1274M G2906S	32	0.000004 0.0001	None	Three affected children, early onset (neonatal, 7 years and 17 months)
Durkie 2021 [11]	S3037L	3	0.000006	S3037L	3	0.000006	None	Prenatal bilateral multicystic kidneys and hydronephrosis. Postnatal: PKD
Durkie 2021 [11]	G960S	4H	0.000009	N3074K	3	0.00003	None	18 months, bilateral multicystic kidneys, atypical, no renal failure
Durkie 2021 [11]	N3188S	4H	0	N3188S	4	0	None, consanguineous parents	Neonatal diagnosis (bilateral), hypertension
Mantovani 2020 [12]	R3277C	3	0.0002	R3277C	3	0.0002	None	22 years, typical ADPKD
Gilbert 2017 [32]	C2495R	4H	0	R3277C	3	0.0002	None	Neonatal onset
Rossetti 2009 [6]	R3277C	3	0.0002	R3277C	3	0.0002	None	62 years, late-onset ADPKD
Durkie 2021 [11]	E3121K	4H	0	R3277C	3	0.0002	None	Prenatal US diagnosis, severe oligohydramnios. Termination of pregnancy
Al-Hamed 2019 [31]	R3938W	3	0.00002	R3938W	3	0.00002	None	Prenatal US diagnosis, oligohydramnios, bilateral polycystic kidneys, hypertension
Durkie 2021 [11]	R3892H	3	0.0005	A3959V	3	0	None	Prenatal US diagnosis, hypertension at birth, enlarged polycystic kidneys
Izzi 2022 [13]	R4154C	2	0.001	R4154C	2	0.001	None	Atypical ADPKD, hypertension, CKD IV at 65 years

**Table 3 genes-14-01230-t003:** Results of the literature review for hypomorphic *PKD1* variants in trans with pathogenic PKD1 variants. Rows are sorted by the codon of the hypomorphic variant, in order to underline the recurrence of the same variant in more than one article (gray boxes). For selection criteria, see “Materials and Methods”.

Reference	Pathogenic Variant	Origin	Hypomorphic Allele	Origin	ACMG Class	H-MAF (GnomAD)	Family History	Clinical Data of the Proband
Audrézet 2016 [9]	T2183fs *	Paternal	D1332N	Maternal	3	0.0002	Father, typical ADPKD	Prenatal US diagnosis (kidneys +3 SD)
Izzi 2022 [13]	Q4231 *	Paternal	D1332N	Maternal	3	0.0002	Father, ESRD 46 years	Enlarged hyperechogenic kidneys in utero, enlarged palpable kidneys at birth, ESRD at 35 years
Gulati 2023 [30]	T2192fs *	Maternal	D1332N	Paternal	3	0.0002	Mother, typical ADPKD	Prenatal US diagnosis, perinatal demise
Gulati 2023 [30]	H526fs *	Paternal	D1332N	Maternal	3	0.0002	Father, typical ADPKD	3 years old, focal cystic disease, hypertension
Gulati 2023 [30]	R2767H	Paternal	V1611I	Maternal	3	0.00003	Some liver cysts in the mother, father unknown but probably ADPKD	20 years old, CKD2, hypertension
Audrézet 2016 [9]	S4169fs *	Maternal	V1611I	Paternal	3	0.00003	Mother, typical ADPKD	Prenatal US diagnosis (kidneys +3 SD, hyperechogenicity)
Durkie 2021 [11]	A1961_Q1962del	Maternal	E1929K	Paternal	3	0.000009	Some maternal relatives affected	Prenatal US diagnosis
Audrézet 2016 [9]	W1839C	Maternal	G1944R	Paternal	3	0.0002	Mother, typical ADPKD	Prenatal US diagnosis (kidneys +2 SD, hyperechogenicity)
Durkie 2021 [11]	S788fs *	Maternal	V1950M	Paternal	3	0.00001	Mother, typical ADPKD	Prenatal US diagnosis, neonatal death
Mantovani 2020 [12]	Y1599 *	Maternal	S2000C	Paternal	3	0	Mother, typical ADPKD	3 years old, focal cystic disease
Audrézet 2016 [9]	N3188D	Paternal	Q2058R	Maternal	3	0.000008	Father, typical ADPKD	At birth, kidneys +14 SD, hyperechogenicity
Gulati 2023 [30]	R3750Q	Paternal	F2132C (1)	Maternal	2	0.0003	Father, typical ADPKD	2 years, ESRD, associated congenital hepatic fibrosis (CHF) and bile duct proliferation
Durkie 2021 [11]	R2266fs *	DN (Pat)	R2162W	Maternal	2	0.0001	No (de novo variant confirmed in trans)	Prenatal echogenic kidneys. Postnatally—multiple bilateral renal cysts identified
Bergmann 2011 [8]	Y2753 *	Paternal	R2255C	Maternal	3	0.00003	Father, typical ADPKD	Earlier onset and more severe disease
Pandita 2019 [33]	c.529 + 3G > C	Maternal	V2267G	Paternal	3	0	Mother affected but with normal renal function	15 years, bilateral polycystosis, dialysis
Bergmann 2011 [8]	R1351fs *	Maternal	L2696R	Paternal	2	0.001	Mother, typical ADPKD	Neonatal diagnosis
Audrézet 2016 [9]	L339fs *	Paternal	T2710N	Maternal	3	0.0004	Father, typical ADPKD	Prenatal US diagnosis (kidneys +3 SD, hyperechogenicity)
Carrera 2016 [34]	V4038fs *	Maternal	V2897delins	Paternal	3	0	Mother, typical ADPKD	Young age (earlier onset)
Durkie 2021 [11]	M1247V	DN (Pat)	R3000C	Maternal	3	0.00002	No (de novo variant confirmed in trans)	Prenatal US diagnosis, postnatal poor renal function, renal transplant aged 7
Durkie 2021 [11]	E2780 *	Paternal	G3150S	Maternal	3	0.000008	Father, typical ADPKD	Prenatal US diagnosis, confirmed bilateral polycystosis on postnatal scan
Mantovani 2020 [12]	L1479fs *	Paternal	I3167F	Maternal	3	0.001	Father, typical ADPKD	Prenatal US diagnosis, two interrupted pregnancies
Durkie 2021 [11]	P252fs *	Paternal	I3167F	Maternal	3	0.001	Father, typical ADPKD	Prenatal US diagnosis reduced amniotic fluid. Postnatal renal impairment and hypertension
Audrézet 2016 [9]	E3872 *	Paternal	R3183Q	Maternal	3	0.0003	Father, typical ADPKD	Prenatal US diagnosis (kidneys +6 SD, hyperechogenicity)
Durkie 2021 [11]	Q1828 *	Maternal	S3238S	Paternal	2	0.00003	Mother, typical ADPKD	2 months, bilateral involvement
Durkie 2021 [11]	R2163 *	Maternal	R3269Q	Paternal	3	0.000006	Mother, typical ADPKD	Prenatal US diagnosis, neonatal demise
Al-Hamed 2019 [31]	L2046P	Paternal	G3227W	Maternal	3	0	Father, typical ADPKD. Single cyst in the mother	Prenatal US diagnosis, postnatal hypertension, enlarged polycystic kidneys
Audrézet 2016 [9]	Gross deletion	Maternal	R3277C	Paternal	3	0.0002	Mother, typical ADPKD	Prenatal US diagnosis, termination of pregnancy
Rossetti 2009 [6]	Q2158 *	Paternal	R3277C	Maternal	3	0.0002	Father, typical ADPKD	Prenatal US diagnosis
Vujic 2010 [7]	R2220W	Maternal	R3277C	Paternal	3	0.0002	Mother, typical ADPKD	Prenatal US diagnosis
Durkie 2021 [11]	C2495R	Maternal	R3277C	Paternal	3	0.0002	Mother, typical ADPKD	Prenatal US diagnosis, severe neonatal PKD, multiple bilateral cysts, hypertension
Audrézet 2016 [9]	W3411 *	Paternal	N3295S	Maternal	3	0.00005	Father, typical ADPKD	Prenatal US diagnosis, hyperechogenicity
Audrézet 2016 [9]	L727P	Paternal	T3945M	Maternal	3	0	Father, typical ADPKD	Prenatal US diagnosis, extremely enlarged kidney, hyperechogenicity, termination of pregnancy
Audrézet 2016 [9]	W861 *	Paternal	E4025G	Maternal	3	0.000004	Father, typical ADPKD	Prenatal US diagnosis, extremely enlarged kidneys, ESRD 2 at 4 years
Bergmann 2011 [8]	L1400fs *	Maternal	R4138H	Paternal	3	0.000004	Mother, typical ADPKD	Two children (neonatal onset and prenatal onset)
Audrézet 2016 [9]	C2370S	Paternal	R4154C	Maternal	2	0.001	Father, typical ADPKD	Prenatal US diagnosis (kidneys +3 SD, hyperechogenicity)
Audrézet 2016 [9]	W1958 *	Maternal	R4154C	Paternal	2	0.001	Mother, typical ADPKD	Prenatal US diagnosis (kidneys +3 SD, hyperechogenicity)

(1) Also in O’Brien et al. 2012 [28] and Lanktree et al. 2019 [29], not meeting selection criteria.

## 4. Discussion

We described two patients who came to our attention for early/very early onset cystic nephropathy and biallelic *PKD1* hypomorphic variants. Both girls had enlarged and echogenic kidneys with onset in utero or soon after birth and unaffected parents. In the absence of a positive family history in either of the parents, these clinical pictures could be misdiagnosed at first for an alternative congenital cystic nephropathy such as Autosomal Recessive Polycystic Kidney Disease (ARPKD), presenting with enlarged echogenic kidneys, hypertension and varying degrees of renal dysfunction due to *PKHD1* or *DZIP1L* biallelic mutations [35], or as *HNF1B*-related disease, where bilateral hyperechogenic kidneys can be detected on prenatal ultrasound [36]. Therefore, these genes were tested in the two probands, ruling out their involvement in both cases. Subsequently, non-truncating, previously unreported biallelic *PKD1* variants were identified, but heterozygous parents did not have renal cysts on ultrasound. It would be interesting to look for minor signs of kidney/liver involvement through abdominal MRI or CT scans, especially in the mother of patient 2, who reported some affected relatives.

These variants were extremely rare in the general population and were classified as Likely Benign, VoUS or Likely Pathogenic according to ACMG standards. Some authors [11] suggested that a different approach could be used while applying ACMG criteria to variants for which a hypomorphic status has been speculated in the literature, for example using the PM3 criteria (recessive in trans) or adding more value to PP5 (reputable source data). As a matter of fact, applying these proposed parameters to the four variants that we reported here contributes to strengthening their pathogenicity. If a diagnosis of bi-allelic ADPKD can be confirmed in these two families, it still leaves many questions unanswered, especially in terms of long-term prognosis and expected age of ESRD, possibility and usefulness of Tolvaptan therapy, and risks related to living kidney donors if the candidate carries a hypomorphic variant. 

In general, the biological effect of hypomorphic *PKD1* alleles is difficult to ascertain, thus making allele-specific clinical case reporting especially valuable [28]. To date, the only variant with unequivocal proven ‘reduced’ function in a genetically engineered mouse mutant (Pkd1 RC mouse) is p.Arg3277Cys, which is also the most frequently reported variant in our literature review. Cellular studies have shown that this amino acid change leads to PC1 misfolding, resulting in increased ER retention and reduced surface expression [37]. Other hypomorphic variants were tested through generated mutant structures in silico by Durkie et al. [11]. In general, the most reliable hypomorphic variants are those reported in multiple affected individuals, such as p.Asp1332Asn, p.Arg4154Cys and again p.Arg3277Cys. Unfortunately, functional studies for *PKD1* variants are cumbersome at the moment, so it is crucial to gather families, obtain long-term follow-up of early onset cases, perform segregation analysis of putative hypomorphic variants and collect population data on variant frequencies. With all the possible limitations, ADPKD serves as a good model for the study of hypomorphic variants, because it is a relatively frequent condition with limited genetic heterogeneity.

The prevalence of hypomorphic variants also has important implications for reproductive genetic counseling. In families where both parents are heterozygous carriers, there will be a 25% recurrence risk in future pregnancies of biallelic ADPKD. In families with typical ADPKD, the risk of bearing a child with early/very early onset of disease can be estimated: the global allele frequency of the hypomorphic variants reported in Table 2 adds up to 1/130; this figure is likely underestimated but would translate into a population frequency of heterozygous carriers of 1/65. The risk is thus relatively low but justifies a warning and consideration of molecular tests in partners with a history of benign cysts.

## 5. Conclusions

PKD1 hypomorphic variants emerge clearly as modifiers of ADPKD, giving a significant contribution to early onset cases. Allele-specific clinical case reporting is extremely valuable in order to increase the recognition of these variants and bring new data related to disease progression, inheritance and response to therapy.

## Figures and Tables

**Figure 1 genes-14-01230-f001:**
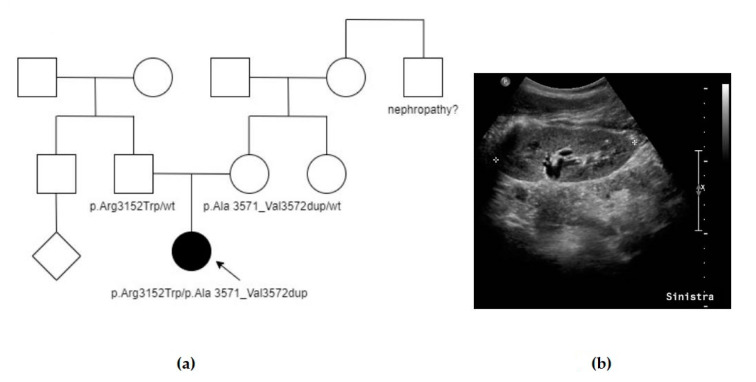
Summary of case F1: (**a**) Pedigree of F1, showing negative family history, except for an uncle with unspecified nephropathy. Affected individuals are colored in black, the arrow indicates the proband. (**b**) Abdominal US scan taken at 12 years old, showing hyperechogenicity with multiple small cysts in the left kidney. Similar signs were seen also in the right kidney (bilateral involvement).

**Figure 2 genes-14-01230-f002:**
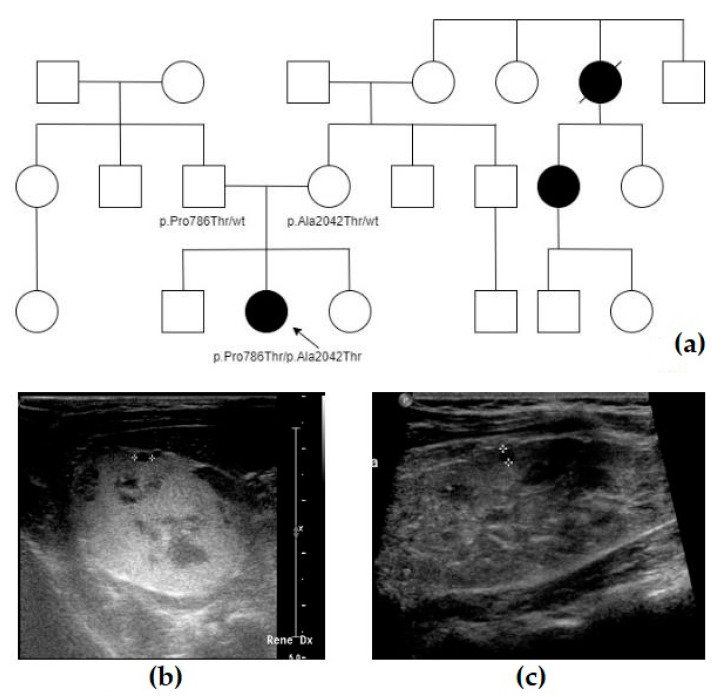
Summary of case F2: (**a**) Pedigree of F2, showing the presence of some relatives on the maternal side with ADPKD. Affected individuals are colored in black, the arrow indicates the proband. (**b**) Abdominal US scan taken right after birth, showing an hyperechogenic right kidney with small multiple cysts. (**c**) Most recent abdominal US scan, taken at 7 years old, showing renal cortical hyperechogenicity with multiple cystic lesions ranging from point-like size to 3–4 mm in diameter.

**Table 1 genes-14-01230-t001:** Genetic variants in *PKD1* gene identified in F1 and F2.

Pt	*PKD1* Variant	Total MAF ^1^	ACMG Criteria	ACMG Class ^2^	Reports ^3^
**F1**	c.10710_10715dup (p.Ala3571_Val3572dup)	0.000005	PM1, PM2, PM4	4	none
	c.9454C > T (p.Arg3152Trp)	0.00001	PM1, PM2, PM5, PP3	4	none
**F2**	c.6124G > A (p.Ala2042Thr)	0.000008	PM1, PM2, PM5, PP3	4	none
	c.2356C > A (p.Pro786Thr)	0	PM2-BP4	2	none

^1^ Minor allele frequency as reported in GnomAD. ^2^ According to VarSome Premium. ^3^ Reports in literature, ClinVar or ADPKD Variants Database (Mayo).

## Data Availability

The data that support the findings of this study are available from the corresponding author upon reasonable request.

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
