# Peer review of "Modifiers of Autosomal Dominant Polycystic Kidney Disease Severity: The Role of PKD1 Hypomorphic Alleles"

_genes, 2023, doi:10.3390/genes14061230_

Round 1
Reviewer 1 Report
This article describes two patients with biallelic PKD1 causing early onset disease. They also review previous descriptions of biallelic PKD1 disease.
Specific Points
1. In families F1 and F2, more sensitive imaging of the parents by MRI or CT would be helpful.
2. Although these seem to be good candidates as hypomorphic PKD1 alleles, it is not clear how 3 of 4 them were scored as LP using the ACMG guidelines. That should be explained. Usually, novel hypomorphic variants score as VUS, which is part if the problem of defining them as disease causing.
3. In the literature review, in families with a negative family history for ADPKD (as in the two families described here) it is not appropriate to always class one variant as main and the other as hypomorphic. If inheritance from the parents, that do not have clear PKD, for the two variants is demonstrated, presumably they are both hypomorphic.
4. It is a little unclear what criteria were used to list families and variants in Table 2. This does not seem to be an exhaustive list from the papers referenced (that does seem quite complete). There may be good reason to exclude some variants, but that rationale should be described more clearly.
5. The study does not provide guidance on how to differentiate a hypomorphic allele from a neutral change, and discussion of this would be helpful. Functional studies?
Minor Points:
1. The first description of PKD1/PKD2 digenic disease is Pei 2001 AJHG
Author Response
We thank the reviewer for these comments and suggestions.
Point 1: In families F1 and F2, more sensitive imaging of the parents by MRI or CT would be helpful.
Response 1: MRI of the parents would surely be helpful in order to identify minor signs of involvement of kidneys and/or liver, but it was not performed. We mentioned this limit in the revised text.
Point 2: Although these seem to be good candidates as hypomorphic PKD1 alleles, it is not clear how 3 of 4 them were scored as LP using the ACMG guidelines. That should be explained. Usually, novel hypomorphic variants score as VUS, which is part if the problem of defining them as disease causing.
Response 2: We added the exact ACMG criteria in the revised Table 1 and described the classification process in the main text. Three variants are classified as LP by VarSome and as VoUS by Franklin: actually, many other supposed hypomorphic variants are scored as LP. Some authors add an “H” to the class (so, class “4H”), like in Durkie et al. 2021.
Point 3: In the literature review, in families with a negative family history for ADPKD (as in the two families described here) it is not appropriate to always class one variant as main and the other as hypomorphic. If inheritance from the parents, that do not have clear PKD, for the two variants is demonstrated, presumably they are both hypomorphic.
Response 3: We thank the reviewer for the suggestion. In the revised text, there are now two distinct tables, table 2 for patients with biallelic hypomorphic variants and table 3 for patients with a clear pathogenic variant (plus one hypomorphic). Hopefully, this will help the readers to better understand our report.
Point 4: It is a little unclear what criteria were used to list families and variants in Table 2. This does not seem to be an exhaustive list from the papers referenced (that does seem quite complete). There may be good reason to exclude some variants, but that rationale should be described more clearly.
Response 4: The exclusion of some families and variants from the medical literature is intentional, following our selection criteria. These criteria are stated in more detail in the revised text and we provide practical examples of exclusion. However, in this revision, we reconsidered some cases and added a few ones.
Point 5: The study does not provide guidance on how to differentiate a hypomorphic allele from a neutral change, and discussion of this would be helpful. Functional studies?
Response 5: This topic is still debated, we addressed this issue in the original text by explaining the relevance of clinical reports and functional studies, which is troublesome in a complex gene like PKD1 (6 pseudogenes, GC-rich regions…etc) and, at the moment, impractical.
Point 6: The first description of PKD1/PKD2 digenic disease is Pei 2001 AJHG
Response 6: We thank the reviewer for this suggestion; we added this suggested reference.
Reviewer 2 Report
I have the following suggestions to strengthen the paper:
For the literature review part, it would be beneficial to include relevant clinical information within the data tables for a more comprehensive overview.
Overall, the quality of the English language used in the paper is satisfactory, but there is room for improvement
Author Response
We thank the reviewer for these comments and suggestions.
Point 1: For the literature review part, it would be beneficial to include relevant clinical information within the data tables for a more comprehensive overview.
Response 1: We added the available clinical data regarding the proband and family (see Table 2 and 3 in the revised text).
Point 2: Overall, the quality of the English language used in the paper is satisfactory, but there is room for improvement
Response 2: We improved English language and corrected several typos.
Reviewer 3 Report
Ambrosini et al. describe two cases of biallelic PKD1 variants in two probands with early onset cystic kidney disease – in this case both diagnosed prenatally. They subsequently perform a literature review of all other previously published cases of biallelic PKD1 variants.
Major Comments:
1. In family 2, the PKD1 p.Pro786Thr variant is poorly conserved evolutionarily and has weak in silico prediction scores that would support benign pathogenicity. Moreover, when I evaluated this variant in Varsome I found it to be classified as likely benign. Could the authors provide evidence to support its pathogenicity? The authors also analyzed only for variants in PKHD1, HNF1B, PKD1, PKD2, and DZIPL1. There are many other genes that can cause early onset cystic kidney disease (e.g. NPHP genes) and therefore demonstrating an absence of pathogenic/likely pathogenic variants in these genes would be important.
2. It would be helpful if the authors could show how they calculated a minimal allele frequency of 1/200 for the hypomorphic variants.
3. It seems rather non-scientific to “arbitrarily” choose a variant as the “main” variant in cases where there are two hypomorphic alleles. Why not include a separate group of probands with two hypomorphic alleles?
Minor Comments:
Page 3 line 133: Can authors clarify what is meant by “multiple spots”? This is not medical terminology that is used in renal ultrasound reports. Do they mean “multiple cysts”?
Page 4 lines 164-165: Some typos: pielic (pelvic?) and urynanalisis (urinalysis)
Overall quality is good. There are many typos (mostly spelling) through the article however that should be addressed (see minor comments above; also several others found that were not listed).
Author Response
We thank the reviewer for these comments and suggestions.
Point 1: In family 2, the PKD1 p.Pro786Thr variant is poorly conserved evolutionarily and has weak in silico prediction scores that would support benign pathogenicity. Moreover, when I evaluated this variant in Varsome I found it to be classified as likely benign. Could the authors provide evidence to support its pathogenicity? The authors also analyzed only for variants in PKHD1, HNF1B, PKD1, PKD2, and DZIPL1. There are many other genes that can cause early onset cystic kidney disease (e.g. NPHP genes) and therefore demonstrating an absence of pathogenic/likely pathogenic variants in these genes would be important.
Response 1: The mentioned variant is classified as LB by VarSome and as VoUS by Franklin. It is not uncommon for putative hypomorphic PKD1 variants to be scored as LB, also considering that functional studies are very limited with a complex gene like PKD1. We inserted the parameters in the revised Table 1. Regarding the other mentioned genes, we actually analyzed the main nephronophthisis genes in Case 2 and we added this piece of information in the revised text. These genes were not tested in Case 1, since PKD1 variants were stronger and clinical/radiological data were less suggestive of nephronophthisis.
Point 2: It would be helpful if the authors could show how they calculated a minimal allele frequency of 1/200 for the hypomorphic variants.
Response 2: We thank the reviewer for this suggestion, we added an explanation in the revised text. This figure is a sum of the MAFs of every single variant reported in Table 2 and Table 3 (a sum of probabilities). Recurrent variants were obviously counted once. By reconsidering MAFs of hypomorphic variants that were initially reported as "main variants", the global figure now adds up to 1/130.
Point 3: It seems rather non-scientific to “arbitrarily” choose a variant as the “main” variant in cases where there are two hypomorphic alleles. Why not include a separate group of probands with two hypomorphic alleles?
Response 3: We provide two separate tables in the revised text, one for patients with a clear pathogenic variant (plus one hypomorphic) and one for biallelic hypomorphic variants, as suggested by the reviewer.
Point 4: Can authors clarify what is meant by “multiple spots”? This is not medical terminology that is used in renal ultrasound reports. Do they mean “multiple cysts”?
Response 4: The term was substituted with “multiple small cysts”.
Point 6: Overall quality is good. There are many typos (mostly spelling) through the article however that should be addressed (see minor comments above; also several others found that were not listed).
Response 6: We improved English language and corrected several typos.
Round 2
Reviewer 3 Report
The authors have addressed my concerns.
Some minor typos can still be found (e.g. in table 2).